# Solid State Fermentation of Shrimp Shell Waste Using *Pseudonocardia carboxydivorans* 18A13O1 to Produce Bioactive Metabolites

**Andi Setiawan** [1], **Widyastuti Widyastuti** [1], **Arik Irawan** [1], **Oklis Syahrin Wijaya** [1], **Aspita Laila** [1], **Wawan Abdullah Setiawan** [2], **Ni Luh Gede Ratna Juliasih** [1], **Kenichi Nonaka** [3], **Masayoshi Arai** [4] and **John Hendri** [1,*]

1   Department of Chemistry, Faculty of Mathematics and Natural Sciences, Lampung University, Bandar Lampung 35145, Indonesia; andi.setiawan@fmipa.unila.ac.id (A.S.); widyadindawione@gmail.com (W.W.); arik.irawaan@gmail.com (A.I.); oklis.syahrin@gmail.com (O.S.W.); aspita.laila@fmipa.unila.ac.id (A.L.); niluhratna.juliasih@fmipa.unila.ac.id (N.L.G.R.J.)
2   Department of Biology, Faculty of Mathematics and Natural Science, Lampung University, Bandar Lampung 35145, Indonesia; wawan.as@fmipa.unila.ac.id
3   Ōmura Satoshi Memorial Institute, Kitasato University, 5-9-1 Shirokane Minato-ku, Tokyo 108-8641, Japan; ken@lisci.kitasato-u.ac.jp
4   Graduate School of Pharmaceutical Sciences, Osaka University, 1-6 Yamada-oka, Suita 565-0871, Japan; araim@phs.osaka-u.ac.jp
*   Correspondence: john.hendri@fmipa.unila.ac.id; Tel.: +62-812-7927-379

**Abstract:** Marine actinomycetes are prolific microorganisms; however, knowledge of their diversity, distribution, and secondary metabolites is limited. Marine actinomycetes represent an untapped source of novel bioactive compounds. In this study, we investigated shrimp shell as substrates for model production bioactive metabolites from actinomycetes under solid state fermentation (SSF) conditions. A total of fifteen actinomycetes were isolated from six sponges and one tunicate. The isolated actinomycetes were grown on solid shrimp shells. Cultures of actinomycetes were extracted with ethyl acetate (EtOAc) and extracts were bioassayed for activity against *Staphylococcus aureus*. One isolate 18A13O1 from the sponge, *Rhabdastrella globostellata*, exhibited antibacterial activity on primary screening compared to the other samples and was chosen for further study. Visualization using SEM showed aerial and substrate mycelia. Through phylogenetic analysis, it was confirmed that isolate 18A13O1 is a *Pseudonocardia carboxydivorans*. Purification of an EtOAc extract yielded A13B2, which showed a minimum inhibition concentration against *S. aureus* at 15.6 µg/mL. It can be concluded that this basic information is very important for further studies related to the development of the production of bioactive secondary metabolites through the solid state fermentation process.

**Keywords:** actinomycetes; antibacteria; *Staphylococcus aureus*; solid state cultivation

## 1. Introduction

The increase in shrimp shell waste from the seafood processing industry has occurred dramatically in recent years [1]. In fact, its economic value is very low, so it often becomes a problem in the collection, disposal, and pollution of waste. Efforts to utilize chitin from shrimp shell waste through chemical processes can have an impact on environmental damage [2]. Taking this into account, there is a need to treat and utilize waste in the most efficient way. The application of solid fermentation technology is the most common biological conversion process for low-cost raw materials, such as shrimp shell waste from the seafood industry [3]. Through this process, microorganisms have the ability to recycle shrimp shell waste to produce high-value compounds to meet industrial needs, while increasing economic value that is sustainable and environmentally friendly. The utilization of shrimp shell waste with an alternative technology to replace chemical methods is a

promising effort in the future to obtain chitin and its derivatives, enzymes, and secondary metabolites.

Recently, solid state fermentation (SSF) technology has received a lot of attention to add value for sustainable products. Although the SSF method is not new, the technology is simple and precise, due to locally available resources such as shrimp shell waste. Chitinolytic microorganisms, one of which is actinomycetes, can decompose shrimp shells. In actinomycetes, chitinase is important in feeding and parasitism. SSF has been applied for the production of secondary metabolites, such as antibiotics [4].

To the best of our knowledge, there is limited information on the use of shrimp shell as substrate fermentation actinomycetes. The members of actinomycetes have been characterized as the most important group of microorganisms in the field of biotechnology, as producers of bioactive secondary metabolites [5]. Marine rare actinomycetes represent a rather untapped source of chemically diverse secondary metabolites and novel bioactive compounds. The results of a study reported from mid-2013 to 2017 showed that there were 97 new species belonging to 27 different rare actinomycete genera, 9 of which were new genera. In addition, the families *Pseudonocardiaceae*, *Demequinaceae*, *Micromonosporaceae*, and *Nocardioidaceae* were the most common, often isolated from the marine environment [6].

Therefore, the purpose of the present study was to confirm the suitability of shrimp shell as a substrate for marine actinomycetes in creating added value to novel products using SSF. The products obtained after SSF were assessed for antibacterial activity. Moreover, the microscopy morphology actinomycetes were evaluated.

## 2. Materials and Methods

### 2.1. Sample Collection and Isolation of Actinomycetes

#### 2.1.1. Sponge Collection

Marine organism samples for isolation of actinomycetes were collected from Singaraja, Buleleng, Bali, Indonesia in August 2018. Samples were collected along the sea coast by SCUBA diving. Collected sponges were identified by their morphological appearance and spicule structures. For spicule identification, small pieces of specimens were placed in Eppendorf tubes and 500 µL of fuming nitric acid ($HNO_3$) were added. Then, samples were mixed with ultrasonic agitation until all organic material had dissolved. Spicules were collected by centrifugation and washed several times with $dH_2O$ [7].

#### 2.1.2. Isolation of Actinomycetes from Marine Sponges

A small piece of sponge was rinsed and homogenized in sterile seawater. The homogenate was submitted to serial dilution and spread on plates of colloidal chitin agar [8] prepared with 50% (*v/v*) artificial seawater (ASW) and supplemented with 25 µg/mL cycloheximide and 25 µg/mL nalidixic acid, and cultured at 28 °C for 14 days. Actinomycetes were isolated and purified on shrimp shell media nourished with 1% colloidal chitin in 50% (*v/v*) agar/seawater. The purified isolate was streaked on slants of 1% colloidal chitin agar at 4 °C and in glycerol 20% (*v/v*) suspensions at −20 °C.

### 2.2. Screening Antibacterial Activity

#### 2.2.1. Clinical Pathogenic Bacteria

The clinical pathogenic *Staphylococcus aureus* used in this study was collected from Abdul Muluk Hospital, Bandar Lampung, Indonesia. Disk diffusion method [9] was used to determine antibiotic resistance patterns in bacteria with some modifications. According to the CLSI guidelines [10], 8 commercial antibiotics were used; clindamycin (2 µg), doxycycline hyclate (30 µg), ciprofloxacin (5µg), cefadroxil (30 µg), lincomycin (2 µg), amoxicillin (25 µg), amoxicillin-clavulanate (30 µg), and erythromycin (15 µg). *S. aureus* was cultured on 2% (*w/v*) nutrient agar (NA). Inoculum was adjusted to 0.5 McFarland standard turbidity (OD 0.08–0.1). After 18 h of incubation, the zone of inhibition for each antibiotic was observed. The results showed *S. aureus* was resistant to amoxicillin-clavulanate (AMX-CL)

in the intermediate category. *S. aureus* was resistant to clindamycin (CDM), ciprofloxacin (CFX), erythromycin (ERH), lincomycin (LC), and amoxicillin (AMX).

### 2.2.2. Resazurin Assay

The assay was done in a sterile 96-well microplate, using resazurin as color indicator [11], and using the broth microdilution method defined by the CLSI. Briefly, serial two-fold dilutions of the extract were prepared by dissolving 2 mg of actinomycetes extract in 1 mL of 12.5% methanol (range, 500 µg/mL to 3.9 µg/mL) and 100 µL of extract solution and 100 µL of bacterial suspension added to each well. The bacterial suspension was prepared from 12 h pure *S. aureus*. Suspensions were adjusted to 0.5 McFarland standard turbidity ($10^6$ CFU/mL) and subsequently incubated for 18–24 h at 37 °C. Wells with 12.5% methanol (MeOH) were used as a solvent control and wells without bacteria were used as a contamination control. An extract control also was included. The lowest concentration of extract that inhibited bacterial growth was taken as the MIC (minimum inhibition concentration) value, as measured with a Hospitex Diagnostic reader. Each test was done in triplicate.

### 2.3. Characterization of Selected Actinomycetes

### 2.3.1. Analysis Morphology

The arrangement of spore and sporulating structures was examined microscopically using the coverslip culture method [12]. A sterile coverslip was inserted at an angle of 45° in the center of 2% chitin agar plate. A loop full of the culture taken from a seven-day-old culture plate was inoculated at the insertion place. After 7 days of incubation at 30 °C, the coverslip was removed and placed upwards on a clean glass slide. The coverslip was finally observed under the light microscope (100×), using an Observer A1 Zeiss microscope.

### 2.3.2. Scanning Electron Microscope

Scanning electron microscopy (SEM) was performed to study the mycelium and spore arrangement of isolated actinomycetes. Stock cultures of actinomycetes were inoculated into 20 mL of a liquid containing 1% *w/v* colloidal chitin diluted with artificial seawater (ASW) in a 100 mL Erlenmeyer flask at 32 °C and static. After 7 days, the shrimp shells were put into a petri dish, as much as 1 g, and 1 mL of bacterial inoculum was added and shaken to moisten the shrimp shells. The culture was incubated for 2–6 days at 32 °C and static conditions. After incubation, a small portion of the shrimp shell was cut using an SLEE Disposable Blades microtome to obtain pieces measuring 0.5 cm × 0.5 cm with a thickness of 0.1 cm. The prepared samples were placed on aluminum stubs, which were fixed with carbon adhesive tabs. The top surface of each stub was then coated under vacuum with a gold layer. The gold plating process was complete in 20 min. Gold plated metal stub was observed on SEM with 10 kV electron high voltage, Carl Zeiss EVO MA 10, Oberkochen, Germany.

### 2.3.3. Phylogenic Analysis of Isolate 18A13O1

Genomic DNA was extracted, following genomic Wizard® Genomic DNA KIT protocol (cat. no. A1120, Promega, Madison, WI, USA). PCR of 16s rDNA sequences was completed using a Sensoquest Sensodirect thermocycler from Germany. PCR was performed using a forward primer: 5′-AGA GTT TGA TCM TGG CTC AG-3′ [13] and a reverse primer: 5′-CCG TAC TCC CCA GGC GGG G-3′ [14], which amplified 848 bp. PCR reaction was completed using 2G Fast ReadyMix Kit (cat. no. KK5102, Merck, Taufkirchen, Germany). The PCR reactions were carried out at a total volume of 25 µL containing 5 µL DNA template (50 ng/L), 12.5 µL 2G Fast ReadyMix, and 6.5 µL RNAse-free water, 0.5 µL forward primer, and 0.5 µL of reverse primer. Amplification was carried out with 35 cycles as follows, denaturation for 60 s at 92 °C, primer annealing for 60 s at 54 °C, and polymerization for 90 s at 72 °C. The PCR results were electrophoresed following the protocol of the QIAXCEL ADVANCED apparatus, Qiagen, Hilden Germany. Sequencing

of PCR results that produce amplicon was carried out using the Sanger method. The results of the sequencing were analyzed phylogenetically using Mega version X software.

### 2.4. Solid State Fermentation

For fermentation, fresh shrimp shell waste (SSW) was obtained from the Lempasing free market, Bandar Lampung. Shrimp shells were washed with tap water at a ratio of 1:10 ($w/v$) twice, using a filter to remove unwanted materials such as soil and sand particles, and dried at $55 \pm 2$ °C overnight in a drying oven. The dried SSW were milled with an electric wearing blender to get shrimp shell chips, and were used as solid SSW substrate for SSF without any further demineralization or deproteinization treatment. The stock bacterial culture was inoculated into 200 mL of liquid contained with 2% *w/v* colloidal chitin diluted with artificial sea water (ASW) in 1000 mL Erlenmeyer conical flask, and was incubated 7 days under 32 °C and static conditions. After this, the concentration of actinomycetes was adjusted to $5.3 \times 10^6$ CFU/mL. Two hundred grams of the substrate was taken in 2000 mL Erlenmeyer conical flask and 200 mL of bacterial inoculum was added and shook to moisten the medium, the culture was incubated for 14 days under 32 °C and static conditions [15].

### 2.5. Extraction, Purification, and Characterization

Actinomycetes biomass was harvested using ethyl acetate (EtOAc) extraction. Filtrate of the EtOAc extract was concentrated by evaporation under reduced pressure. The EtOAc extract was partitioned into a water–EtOAc mixture (1:1). The active fraction was fractionated using $SiO_2$ gel open column chromatography and then purified using a Buchi Sepacore X50 system with a glass $C_{18}$ column (in dia.15 mm × L. 230 mm). Purification was based on bioassay guided separation. Final active fraction was characterized by Thermo Scientific™ Dionex™ Ultimate 3000 RSLCnano UHPLC coupled with Thermo Scientific™ Q Exactive™ High Resolution Mass.

## 3. Results and Discussions

### 3.1. Sample Collection and Isolation of Actinomycetes

Actinomycetes were isolated from marine organisms collected at Buleleng, Bali, Indonesia. During a one-week collection trip, 65 marine organisms were collected along the coast (8°07′20.9″ S 114°34′03.8″ E) at a depth of 5–20 m. Based on morphology and appearance, this indicated that most of the samples collected were sponges of the genus *Desmospongiae*, which were characterized by siliceous spicules [16]. In the course of our study, a total of fifteen actinomycetes isolates were isolated from six sponges (18A13, 18B18, 18D33, 18D35, 18E41, 18F47) and one tunicate (18D36), as showed in Figure 1. Isolation of actinomycetes from varied marine organisms has also been reported by researchers for identifying producers of bioactive metabolites [17,18].

### 3.2. Screening Antibacterial Activity

All isolated actinomycetes were cultivated in shrimp shell media and screened against *S. aureus*, which is resistant to clindamycin (CDM), ciprofloxacin (CFX), erythromycin (ERH), lincomycin (LC), and amoxicillin (AMX). As showed in Table 1, the EtOAc extract of isolate 18A13O1 exhibited the strongest inhibition of *S. aureus* growth at a concentration of 250 μg/mL.

Actinomycetes 18A13O1 was isolated together with 18A13A1 from sponge 18A13. Sponge 18A13 (Figure 2) was shaped like a ball with reddish-purple exterior color, and yellowish internal color. This sponge has long tubular spicules with two pointed ends, characteristic of *Demospongiae* [19]. It was identified as *Rhabdastrella globostellata*, which is common along the Indonesian coast. The results of previous studies which show that *R. globostellata* produce isomarabaric compounds and have antiproliferative activity against vascular endothelial cells [20]. It was recently reported that this type of sponge produces bioactive compounds with anti-leukemic activity [21]. However, studies of bioactive

compounds from microorganisms, especially actinomycetes associated with *R. globostellata*, are still scarce.

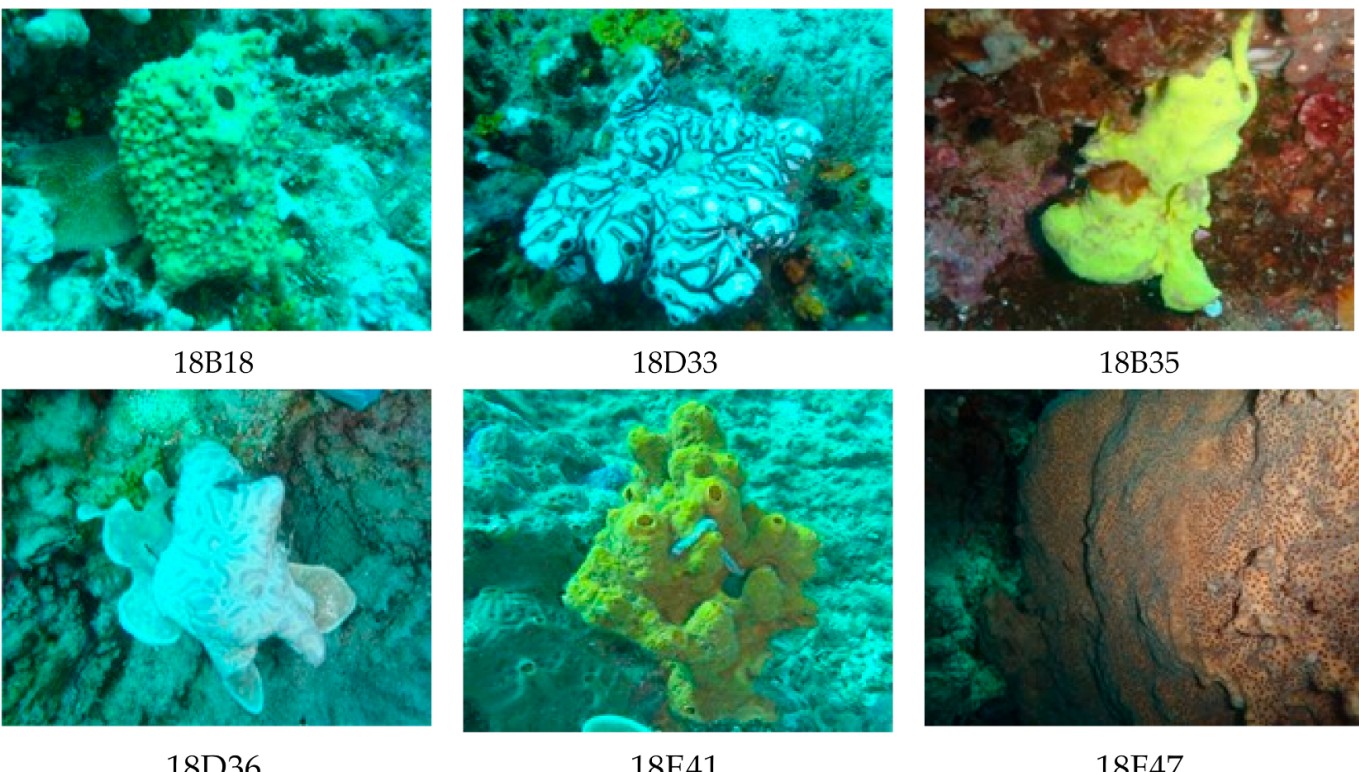

**Figure 1.** Collection of some marine organisms from Buleleng, Bali.

**Table 1.** Screening actinomycetes EtOAc extracts.

| No. | Sample Code | Phylum | Isolate Actinomycetes | Color | Inhibition Concentration (μg/mL) |
|---|---|---|---|---|---|
| 1 | 18A13 | Porifera | 18A13A1 | White | 500 |
|   |       |          | 18A13O1 | White | 250 |
| 2 | 18B18 | Porifera | 18B18A1 | White | 500 |
|   |       |          | 18B18A2 | White | 500 |
|   |       |          | 18B18A3 | White | - |
|   |       |          | 18B18A4 | White | 500 |
| 3 | 18D33 | Porifera | 18D33A1 | White | - |
|   |       |          | 18D33A2 | White | 500 |
| 4 | 18D35 | Porifera | 18D35A1 | Grey | 500 |
|   |       |          | 18D35A2 | White | - |
| 5 | 18D36 | Tunicate | 18D36A1 | Grey | 500 |
|   |       |          | 18D36A2 | Grey | - |
| 6 | 18E41 | Porifera | 18E41A1 | White | 500 |
| 7 | 18F47 | Porifera | 18F47A1 | White | - |
|   |       |          | 18F47A2 | White | 500 |

(-): inactive.

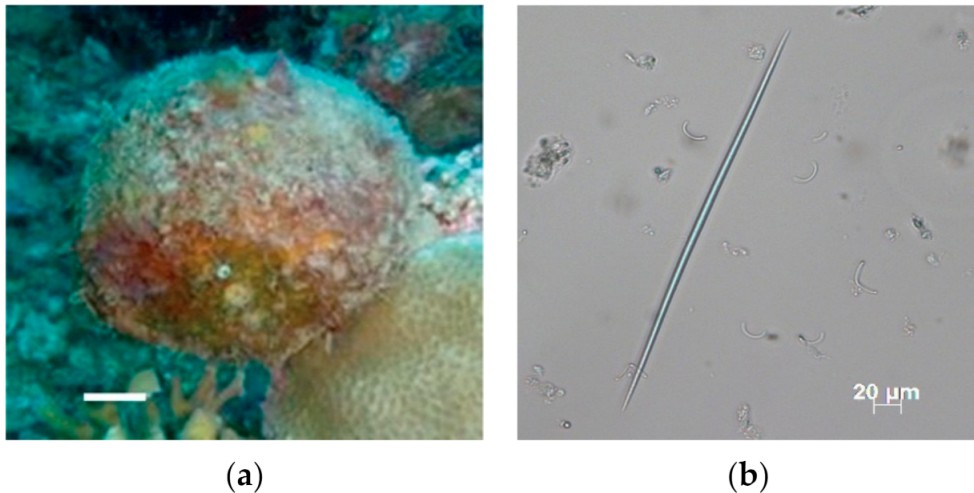

(**a**)  (**b**)

**Figure 2.** (**a**) Sponge 18A13 bar 3 cm, (**b**) spicule of the same specimen.

### 3.3. Characterization of Selected Actinomycetes

A visual observation of actinomycetes cultivated in media showed mycelia penetrating the substrate, while aerial *mycelia* grew vertically at the media-air interface. Under microscopic observation, all isolated actinomycetes had a mycelium that was about 2 microns wide and the diameter of the spores was around 6.1 microns [22]. This is smaller than many fungi which have *mycelia* of 2–30 microns [23]. Actinomycetes are anaerobic microorganisms. They show filamentous and branching growth patterns on solid substrates resembling fungi mycelia. Their colonies are extensive like mycelium. Aerial hyphae are found in many genera of actinomycetes. As shown in Figure 3c, which suggests isolate 18A13O1 as actinomycetes.

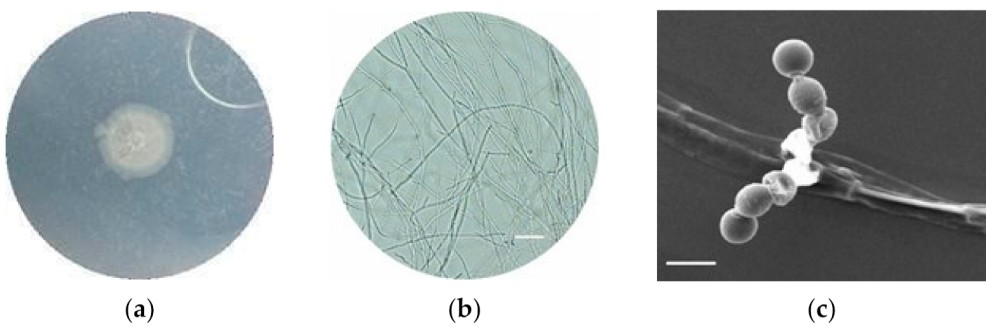

(**a**)  (**b**)  (**c**)

**Figure 3.** (**a**). Isolate 18A13O1 in colloidal chitin media, (**b**) visualization 18A13O1, bar 40 micron with light microscope scale 200×, (**c**) SEM image aerial hyphae isolate 18A13O1, bar 4 micron.

### 3.4. Phylogenetic Analysis of Isolate 18A13O1

Actinomycete isolates collected from the marine sponge, *R. globostellata*, were investigated. Isolate 18A03O1 was selected based on the bioactivity of the extract. The 16S rDNA gene was sequenced, and the resulting sequences were blasted against the GenBank database. Isolate 18A03O1 was stated to belong to the genus *Pseudonocardia*. The new strain *Pseudonocardia carboxydivorans* 18A03O1 was identified based on sequence similarity of 99.65% and was registered in the gene bank with access number LC647653. Through phylogenetic analysis, it was found that the actinomycetes isolate 18A13O1 was *P. carboxydivorans* (Figure 4).

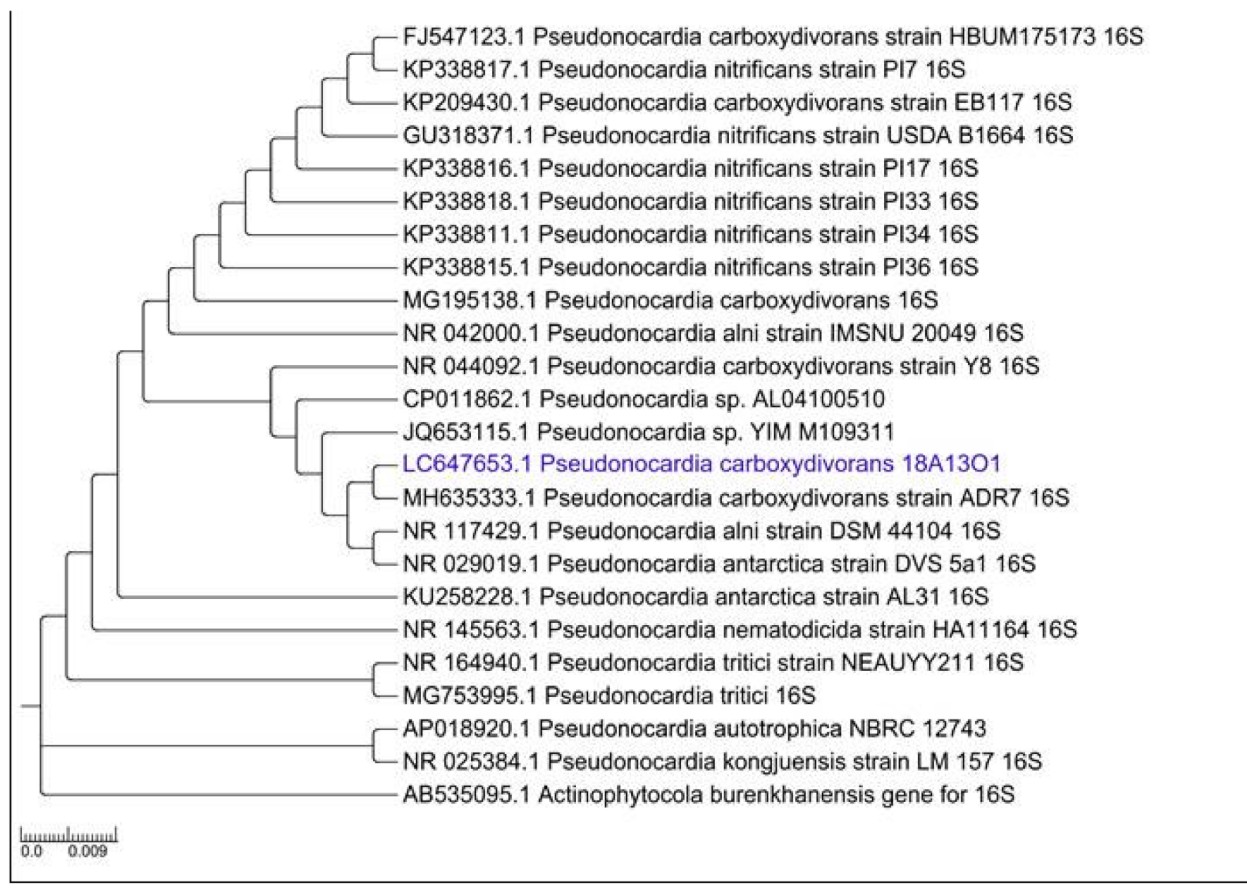

**Figure 4.** Phylogenetic tree using maximum likelihood method and Tamura 3-parameter model of 22 *Pseudonocardia* representatives and *Actinophytocola burenkhanensis* (Acc. No. AB535095) as an outgroup. Bootstrap values (1000 resamples) are given in percent at the nodes of the tree. The isolate *P. carboxydivorans* 18A03O1 is presented in bold.

Based on the literature review, *P. carboxydivorans* was included in the rare marine actinomycetes [24]. Furthermore, it was reported that through the fermentation process, *P. carboxydivorans* was able to produce Branimycin B and C macrolides, which have antibacterial activity [25]. However, the bioactivity of the metabolites of *P. carboxydivorans* 18A03OI isolated from solid fermentation using shrimp shell media has never been reported.

### 3.5. Solid State Fermentation

In order to isolate bioactive metabolites, actinomycetes were cultivated in solid, shrimp shell media. (Figure 5a). Actinomycetes appeared at the surface of the shrimp shell, exhibiting a white color after 7 days.

After two weeks, the inoculum was spread across the surface of the shrimp shell media and cultivated for 21 days (Figure 5b). In total, 800 g of shrimp shells was used to cultivate isolate 18A13O1. Solid state fermentation (SSF) was accomplished in a 1 L Erlenmeyer flask which contained 200 g of shrimp shells. Actinomycetes growth on the surface of shrimp shell media were characterized by the formation of aerial and substrate mycelia (Figure 5c) [26]. Figure 5 shows actinomycetes can grow well on shrimp shells. However, information on bioactive compounds from actinomycetes grown on shrimp shells is still very limited [27]. Isolate 18A13O1 showed characteristic spores with ball-like shapes. After three weeks, actinomycetes biomass was extracted with EtOAc and evaporated under reduced pressure to obtain a crude extract EtOAc (1.4 g).

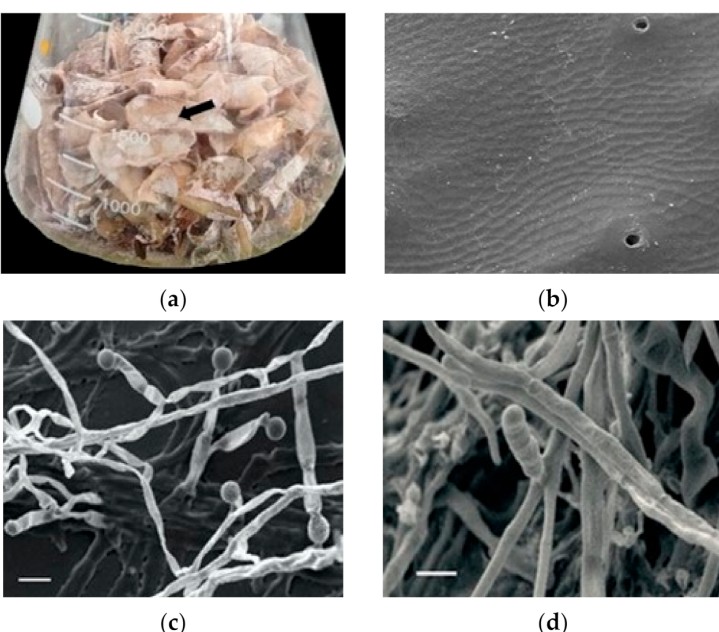

**Figure 5.** SEM image (**a**) solid state fermentation 18A13O1 in shrimp shell media, (**b**) surface shrimp shell, (**c**) actinomycetes 18A13O1 on solid shrimp shell media after 7 days bar 20 μιχρον, and (**d**) after 21 days 20 micron.

### 3.6. Extraction, Isolation, and Characterization

An EtOAc extract of 18A13O1 (1.1 g) that exhibited antibacterial activity against *S. aureus* at 250 μg/mL was partitioned into a water-AcOEt mixture to obtain an AcOEt soluble portion (772 mg). The AcOEt soluble portion showed growth inhibition at 62.5 μg/mL against *S. aureus.* This fraction was subjected to $SiO_2$ gel open column chromatography eluted with n-hexane and isopropanol, affording two fractions: 147 mg A13A1 (eluted with n-hexane 100%) and 358 mg A13A2 (eluted with n-hexane:isopropanol 3:1). The active fraction, A13A2, was further purified using a Buchi Sepacore X50 system with a glass column $C_{18}$ Nacalai (linear gradient $H_2O$:MeOH; MeOH 30% to 90%) into five fractions A13B1 (18 mg), A13B2 (90 mg), A13B3 (41 mg), A13B4 (37 mg), A13B5 (22 mg). Each fraction was subjected to bioactivity analysis and A13B2 inhibited growth of *S. aureus* most effectively at a concentration of 15.6 μg/mL. Analysis using HRLCMS showed molecular ion peak $[M+H]^+$ at *m/z* 441.37164 in Figure 6. This spectrum indicated that the active compound A13B2 was analog to Branimycin B which has one less double bond. In general, specialized metabolites are often restricted to a constricted set of species within a phylogenetic group [28].

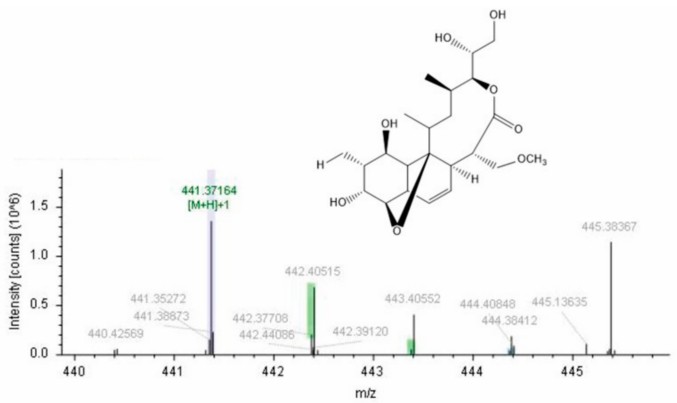

**Figure 6.** HRLMS spectrum of fraction A13B12, $C_{23}H_{36}O_8$, $[M+H]^+$ = m/z 441.37164.



## 4. Conclusions

To our knowledge, this is the first report of solid state fermentation of marine sponge-associated *P. carboxydivorans* 18A13O1 on shrimp shell media. This study leads to an understanding of the potential of microorganisms producing bioactive compounds through solid state fermentation processes. Based on this research, actinomycetes derived from sponges can grow well on selective media of shrimp shell waste in the fermentation process to produce antibacterial compounds. This basic information is very important for further research related to the development of bioactive compounds through the solid fermentation process.

**Author Contributions:** J.H., A.S. and M.A. conceived and designed the experiments; A.S., W.W., A.I., O.S.W., A.L., W.A.S., K.N. and N.L.G.R.J. performed the experiments; all authors analyzed the data; J.H., A.S. and M.A. wrote and revised the manuscript. All authors have read and agreed to the published version of the manuscript.

**Funding:** This research was funded by The Deputy of Research Strengthening and Development, The Ministry of Research and Technology/National Agency for Research and Innovation of the Republic of Indonesia, Basic Research with grand no. 139/SP2H/ADM/LT/DRPM/2020, dates, July, 2020 to JH. This research was also partially supported by the Platform Project for Supporting Drug Discovery and Life Science Research (Basis for Supporting Innovative Drug Discovery and Life Science Research (BINDS)) from the Japan Agency for Medical Research and Development (AMED) (grant no. JP21am0101084), the Kobayashi International Scholarship Foundation, and a Grant-in-Aid for Scientific Research B (grant nos. 18H02096, 17H04645 and 21H02069) from the Japan Society for the Promotion of Science (JSPS) to MA.

**Institutional Review Board Statement:** Not applicable.

**Informed Consent Statement:** Not applicable.

**Data Availability Statement:** Not applicable.

**Acknowledgments:** The authors acknowledge the Directorate of Resources, Directorate General of Higher Education, Ministry of Education, Culture, Research, and the Technology, Republic of Indonesia, for their continuous support in WCP program and to all laboratory staff members at UPT Laboratorium Terpadu Sentra Inovasi dan Teknologi, Universitas Lampung, for supporting the laboratory facilities.

**Conflicts of Interest:** The authors declare no conflict of interest.

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
