# Peer review of "Solid State Fermentation of Shrimp Shell Waste Using Pseudonocardia carboxydivorans 18A13O1 to Produce Bioactive Metabolites"

_fermentation, doi:10.3390/fermentation7040247_

Round 1

Reviewer 1 Report

The authors added material on DNA identification, on the basis of which they confirmed the species belonging of one of the strains. There are two forward primers listed in the Methods section, this is probably a mistake. Please confirm routinely which primers for the 16S rRNA gene analysis you used (F and R). Reference [13] does not contain a primer to which the authors refer.

After this clarification, the work can be published.

Author Response

Dear Reviewer 1,

Thank you very much for your comments.  

We attached the responds of your comments.

We really appreciate about your suggestions to improve the quality of 

the manuscript. 

Reviewer 2 Report

The paper is well written and of moderate interest. However, I feel confused about the microorganism considered in the study. The microorganism is reported to be Pseudonocardia but the pictures (see Figure 5) depict probably a fungus and not an actinomycete. Also the dimensions reported for the microorganism are compatible with fungi. If chemical extraction was performed exactly on shells of figure 5, then the producer of the antibiotic compound should be the fungus (eventually could even be e mixed culture). If not, the authors should provide the original picture of that fermentation. I strongly suggest that the authors address to an expert in classification of microorganisms in order to get rid of this doubt. Note that 16S analysis will reveal bacteria even if they are present as contaminants but will not reveal fungi.

Author Response

Dear Reviewer 2

Thank you very much for your comments. 

We responds your comments as attachment file.

We are really appreciate for your suggestions to improve

the quality of the manuscript.

Best regards,

John Hendri

Reviewer 3 Report

Chapter Results – change in Results and Discussions

Chapter 5 – Conclusions is Chapter 4

Author Response

Dear Reviewer 3, 

Thank you for your suggestions and comments regarding to our manuscript. 

As your suggestions, we changed chapter results become results and discussions, and chapter 5 to be chapter 4.

We really appreciate your suggestions and comments.

Best regards,

John Hendri

Round 2

Reviewer 2 Report

comment to question 1.

If the picture is the actual picture of the culture at the base of the work, the authors have probably not worked with Pseudonocardia but with a fungus

Comment to question 2.

In the absence of the feedback of an expert in fungi. I cannot accept a self-certification. I understand that the 16s data have identified a Pseudonocardia but this is not sufficient to certify that all the job was done by a Pseudonocardia. Note also that Pseudonocardia could install symbiosis with fungi, and the 16s result could therefore derive from symbiotic/contaminant Pseudonocardia in the culture. Additional tests must be done for fungi (18s)